# Executable but Unlearnable: Designing Code That Resists LLM-Based Learning

**Viraaji Mothukuri**
Kennesaw State University
Marietta, USA
vmothuku@students.kennesaw.edu

**Reza M. Parizi**
Kennesaw State University
Marietta, USA
rparizi1@kennesaw.edu

## Abstract

The LLM foundation model era has inverted a fundamental assumption in software engineering. Code, once written, no longer belongs exclusively to its creators. Any publicly accessible code becomes training data, absorbed into models that can reproduce, adapt, and redistribute it without consent. This paper argues that such circumstances represent not merely a legal or ethical challenge, but a *technical* one requiring new defensive primitives. Without technical defenses, the most valuable software becomes training data for systems its authors do not control. We introduce the concept of *Statistical Opacity*, defined as the deliberate design of code representations that resist neural pattern extraction while preserving human readability and machine executability, by exploiting the gap between executability and learnability. We articulate a research agenda spanning theory, pathways, tools, and evaluation.

## CCS Concepts

• **Security and privacy → Software and application security**;
• **Computing methodologies → Natural language processing**.

## Keywords

Statistical Opacity, Unlearnable Code, AI-resistant Software, Foundation Models, Software Security, Large Language Models (LLMs)

**ACM Reference Format:**
Viraaji Mothukuri and Reza M. Parizi. 2026. Executable but Unlearnable: Designing Code That Resists LLM-Based Learning. In *Proceedings of the 3rd ACM International Conference on AI-Powered Software (AIware '26), July 6–7, 2026, Montreal, QC, Canada.* ACM, New York, NY, USA, 6 pages. https://doi.org/10.1145/3805760.3814896

## 1 Introduction

In contemporary software security, the efficiency and reliability of protective mechanisms depend on robust threat models and comprehensive defensive strategies. Traditional approaches assume adversaries seek to *execute* unauthorized operations, leading to defenses built around access controls, input validation, memory safety, and cryptographic protocols. These defenses assume adversaries interact with software through its *interfaces*. Large Language Models (LLMs) such as GPT, Claude, and Gemini invalidate this assumption. These new adversaries do not need to execute code. They only need to *read* it to extract statistical patterns during training. Once

learned, the patterns can be reproduced, adapted, and deployed in ways the original authors never intended and cannot control the usage [20]. This paradigm shift enables attackers to derive capabilities from code without ever running it, fundamentally altering the threat landscape of software protection. The trajectory of this threat is evident in recent developments. LLMs when equipped with a structured abstraction layer, can plan and execute sophisticated multi-host cyberattacks on realistic emulated enterprise networks [10]. By enabling LLMs to specify high-level attack actions that domain-specific agents translate into concrete exploit sequences. Without the abstraction layer, even frontier models succeeded in only 3 of 40 environments, underscoring how close LLMs are to operational offensive capability and how rapidly the remaining scaffolding gap is narrowing.

Google's Threat Intelligence Group (GTIG) identified PROMPT-FLUX and related malware families as the first confirmed instances of malicious code that invoke LLM capabilities during execution [11]. Written in VBScript, PROMPTFLUX queries Gemini's API through a "Thinking Robot" module that requests obfuscation and evasion techniques, establishing a recursive cycle of mutation in which the malware periodically regenerates its own source code. Although GTIG assessed PROMPTFLUX as experimental and not yet capable of compromising a victim network, the report characterized this class of malware as a significant step towards autonomous and adaptive threats. Evaluation across 1,354 test cases from five real-world projects [32] showed that the decrease in test pass rate ranged from 15.3% under symbol-level obfuscation to 62.5% under combined symbol, structure, and semantic transformations. Even at the highest obfuscation intensity, models retained over a third of their code generation capability, and lighter transformations had minimal effect. Another study [6] found that LLMs achieve 97.23% syntactic and 60.93% semantic deobfuscation [30] on JavaScript programs, with GPT-4o and Codestral successfully deobfuscating more programs than traditional baseline approaches.

As organizations increasingly deploy AI-assisted development tools, several critical challenges emerge. Concerns regarding intellectual property protection as proprietary algorithms become training data for foundation models. Inability to maintain security [28] guarantees when attackers can learn vulnerability patterns from publicly available code. Growing evidence that traditional obfuscation provides insufficient protection against neural comprehension. To address these obstacles, we propose Statistical Opacity as a new security primitive. This property ensures that code remains functional for intended users while degrading its utility as training data for machine learning (ML) systems. The approach exploits the gap between what makes code executable (syntactic and semantic correctness) and what makes code learnable (statistical

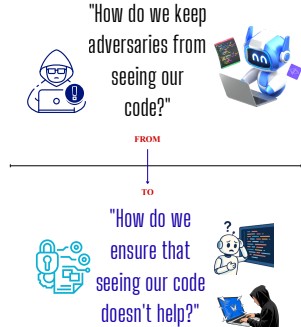

**Figure 1: Security Question Reframed *from* Access Prevention *to* Learnability Resistance**

regularity) (refer to Figure 1). This work provides a comprehensive vision for AI-resistant software, including formal definitions, theoretical foundations, and a research agenda. The **contributions** can be summarized as providing a definition and theoretical grounding for Statistical Opacity as a security primitive, identifying candidate pathways for achieving opacity in code representations, and articulating a research agenda.

## 2 Statistical Opacity

We define **Statistical Opacity** as a property of code representations in which the statistical patterns extractable via gradient-based learning are decoupled from the semantic content required for correct execution. In practical terms, a statistically opaque program *runs correctly* but *teaches badly*. More precisely, let $P$ denote a program with defined input and output semantics, and let $T$ denote a transformation such that $T(P)$ is semantically equivalent to $P$ for all valid inputs. Let $\mathcal{L}$ represent a gradient-based LLM trained on a corpus containing $T(P)$ and let $\mathcal{L}'$ represent a baseline model trained on the same corpus with $P$ in its untransformed form. Statistical Opacity holds when the generalization performance of $\mathcal{L}$ on tasks derived from $P$ degrades measurably relative to $\mathcal{L}'$, where degradation is quantified through standard evaluation benchmarks (HumanEval [7]) for code completion. The transformation $T$ is opacity-conferring to the extent that it preserves execution semantics while maximizing this generalization gap across a defined family of gradient-based learners. Statistical opacity is not a refinement of obfuscation. The two primitives target different adversary classes, optimize different objectives, and admit different impossibility boundaries. Statistical opacity targets a narrower set of adversaries than traditional defenses. Where obfuscation aims to resist any polynomial-time adversary, an impossibly strong guarantee that is formally unattainable in general [4], opacity targets only gradient-based learners. Restricting the adversary class makes the goal tractable.

Three converging trends make this primitive urgent. First, recent research demonstrates that LLMs integrated into hierarchical agent frameworks can serve as autonomous red-team agents capable of coordinating and executing multi-step cyberattacks across realistic enterprise network environments. As these capabilities grow, so

does the value of pattern resistance. Second, foundation model developers have exhausted most publicly available code [31]. Private codebases, proprietary systems, and enterprise software become the next sources. Organizations face a choice between using their code as training data or developing technical countermeasures. Third, the same LLMs that help developers write code can help attackers understand it. Every security-critical system becomes more vulnerable as neural code comprehension improves. We propose a reframing of the security question. The previous formulation asked how to keep adversaries from seeing code. The proposed formulation asks how to ensure that seeing code does not help. This mirrors the conceptual shift in cryptography, where the community stopped trying to hide algorithms and instead designed algorithms that remain secure even when fully known.

## 3 Theoretical Foundations

Prior work [12] established that source code is *more* statistically regular than natural language, exhibiting lower cross-entropy, more predictable patterns, and stronger local regularities. This naturalness makes code amenable to ML. Our observation is that naturalness is a design choice; programmers write naturally by convention, not by computational necessity. If code is written unnaturally while preserving correctness, learnability can be degraded. If code is written unnaturally while preserving correctness, learnability can be degraded. Recent work on the Structured Naturalness Hypothesis [25] extends this finding to abstract syntax trees and control flow graphs, proposing that Zipfian regularities of the kind first established for natural language [16] carry over to syntactic structures of code. This suggests that opacity must operate across lexical, syntactic, and semantic levels.

Research on unlearnable examples [13] established that images can be made unlearnable through imperceptible perturbations that prevent models from extracting useful patterns during training. The perturbations exploit gradient-based learning, inducing shortcuts that override genuine features. The learning process is poisoned without corrupting the data's primary function. Subsequent work [14] adapted this approach to code, using CodeBERT embeddings to guide a multi-armed bandit search over lightweight semantics-preserving transformations. CoProtector [29] demonstrated that untargeted corruption combined with watermark backdoors reduces code model accuracy by 7.3% at a 10% poisoning rate, with watermarks remaining detectable even at 0.1% poisoning.

A theoretical result [5] connects learning and obfuscation as dual operations. If a concept class can be learned with $M$ mistakes, it can be canonically obfuscated in time $O(M \cdot \text{poly}(n))$. Conversely, certain obfuscations are impossible because the dual class is learnable. The duality formalizes why opacity is tractable. Full obfuscation must resist any polynomial-time adversary, while opacity restricts the adversary class to gradient-based learners. A subsequent result [8] showed that PAC learnability implies backdoor defendability but not vice versa, with direct implications for the security of code against poisoning attacks.

## 4 Pathways to Statistical Opacity

Five families of pathways achieve Statistical Opacity, each grounded in how current models process code. The first three target distinct

layers of the learning pipeline directly. The remaining two operate at the level of corpus distribution and runtime behavior. Figure 2 summarizes the dual-output flow that the pathways jointly produce.

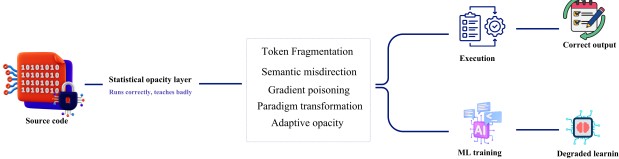

**Figure 2: Statistical Opacity Pathways**

Modern LLMs use fixed tokenization schemes (BPE, WordPiece) [27] trained on general text corpora. Code is tokenized under these same schemes, which creates an exploitable structure. The first pathway, **token fragmentation**, designs identifiers that fragment unpredictably under standard tokenizers. A variable named `calculateTotal` tokenizes predictably into a small number of subwords, while one named `cAlCuLaTe__tOtAl` fragments into many more, since BPE was trained on lowercase text and treats irregular casing as out-of-distribution. Models learn associations between token sequences, so when semantically related code produces divergent token sequences, cross-example generalization fails [2]. Empirical testing across 13 LLMs on 250 Java problems [22] found that variable renaming alone caused an 18.6% accuracy drop, while substitution of string and numeric literals caused a 21.4% drop. These effects compound when multiple opacity layers are applied in sequence.

The second pathway, **semantic misdirection**, constructs a vocabulary where keywords trigger incorrect priors. Models learn that the keyword `for` predicts iteration, `class` predicts object-oriented structure, and `async` predicts concurrency patterns [15]. These associations transfer across codebases and form the bedrock of code completion. An opacity-enhanced representation systematically violates them, presenting collection traversal under the `while` idiom or stateful behavior under nominally functional constructs. The model's keyword priors lead to mispredictions in the transformed code, which propagate through downstream fine-tuning and inference.

The third pathway, **gradient poisoning**, inserts functionally neutral code (dead branches, redundant computations, unreachable paths) crafted to dominate gradient updates during training. The model spends representational capacity on learning patterns that have no predictive value for the function's actual behavior. This constitutes an adversarial attack on the training process itself, analogous to data poisoning [9], but operating via code structure rather than corrupted labels. Because the inserted constructs are syntactically valid and semantically inert, they survive standard preprocessing pipelines. Research on covert poisoning [1] demonstrated 20% attack success rates while evading signature-based cleansing, suggesting that even modest poison rates can shift model behavior.

The fourth pathway, **paradigm transformation**, rewrites code into semantically equivalent forms drawn from paradigms underrepresented in model training data. Code LLMs are trained predominantly on imperative and object-oriented code in mainstream languages. Functional, declarative, point-free, and combinator-based

representations exist in training corpora but at far lower frequency. Transforming a standard imperative loop into a fold-map-filter composition, or converting a stateful class into a closure-based factory, shifts the source out of the model's high-density training distribution. The same algorithm, when rewritten in a low-frequency paradigm, exhibits higher per-token perplexity even while preserving human readability.

The fifth pathway, **adaptive opacity**, treats protected code as a moving target rather than a static artifact. The previous four pathways are applied once at build time and shipped. Adaptive opacity instead varies transformation parameters or applies additional pathways at runtime when adversarial analysis is suspected, drawing on principles from moving-target defense in network security. A library might serve a lightly transformed variant under normal access patterns and switch to a heavily transformed variant when scraping signatures are detected, or vary opacity per-call based on the calling environment. The research agenda treats this pathway as a Phase 4 milestone, since reliable signals for distinguishing legitimate access from adversarial probing remain an active research area.

A defender who knows the technique used in one family cannot use that knowledge to neutralize another, which motivates composing them. Section 7.1 demonstrates the first three pathways applied in composition; paradigm transformation and adaptive opacity remain agenda items, not implemented in the current reference tool. Phase two of the research agenda (Section 5) explicitly calls for discovering additional pathway families beyond the five named here, treating the present taxonomy as a starting point rather than a complete enumeration.

## 5 Research Agenda

We propose a phased research agenda spanning theory, pathways, tools, and evaluation. During the first phase, Statistical Opacity should be formalized by rigorously defining connections among learnability theory, information theory, and program semantics. Establishing baselines requires systematically measuring how existing obfuscation tools affect model learning. The Loki framework [26] serves as the current gold standard for ML-resistant obfuscation, leveraging VM-based protection and formally verified MBA expressions to reduce synthesis success to 19%. Understanding which techniques transfer to neural adversaries is essential. Developing evaluation frameworks requires determining appropriate metrics, such as Pass@k for code completion, accuracy for vulnerability detection, or transfer learning degradation. During the second phase, building source-to-source transpilers requires creating tools that transform standard code into opacity-enhanced representations, with parameterizable, reversible transformations for authorized maintainers. Exploring language-level solutions requires investigating whether certain programming paradigms are inherently more opaque.

During the third phase, IDE and toolchain integration should make opacity as easy to enable as compiler optimization flags. Selective opacity techniques should protect security-critical sections while leaving other code transparent. Opacity-aware development practices should address how teams write code that will later be opacity-transformed. During the fourth phase, red teaming efforts

should develop adversarial training methods, preprocessing defenses, and multi-model ensemble approaches that attempt to break opacity. Adaptive opacity systems should detect when their opacity is being probed and respond dynamically. Formal verification should demonstrate that certain transformations guarantee opacity bounds against specific model classes. We make an initial proof-of-concept implementation publicly available [21].

## 6 Applications and Implications

The most immediate application is protecting proprietary code from unauthorized training. Organizations invest substantial resources in developing unique algorithms, optimizations, and implementations. Statistical Opacity offers a technical complement to legal protections. Code that implements authentication, cryptography, or access control is especially sensitive. If attackers can learn patterns from similar implementations, they can more easily identify vulnerabilities or craft exploits. Recent research demonstrates that LLMs can already autonomously coordinate multi-step attacks. Opacity-protected code is, by design, harder to analyze via learning.

As foundation models become infrastructure, questions of consent and compensation for the use of training data become urgent. Statistical Opacity gives code authors a technical mechanism for withholding their work from training pipelines, independent of legal frameworks that vary by jurisdiction. We acknowledge that Statistical Opacity is dual-use. Techniques that protect legitimate software equally protect malware. GTIG's analysis of PROMPT-FLUX demonstrates that threat actors are already exploring how to make their code resistant to analysis. Cryptography faces the same duality. Statistical Opacity, like cryptography, requires thoughtful development with attention to responsible disclosure and defensive applications.

The hardest constraint is that, unlike images, code cannot tolerate continuous noise; every transformation must preserve syntactic validity and semantic correctness. Models will adapt if opacity techniques become widespread, so defenses must be designed for an adversary that knows the techniques and optimizes against them. Research on deobfuscation pre-training [3] demonstrated that training on deobfuscation yields a 12.2% improvement in code translation tasks, suggesting that obfuscation itself can serve as a training signal. A further challenge is achieving differential opacity, where transformations degrade machine comprehension while preserving human readability. Scaling these techniques from experimental codebases to production systems with millions of lines represents another essential direction.

## 7 Scenarios and Feasibility

Code model capability advances on each model generation, while defense research operates on multi-year cycles. Standard benchmarks for code generation have seen first-attempt accuracy rise from below 30% to above 96% in three years [7, 23, 24], though more rigorous test suites reveal that headline numbers overstate model robustness. If deobfuscation capability tracks a comparable trajectory, pathways designed early in the agenda will face stronger adversaries before later designs are validated. The agenda must

therefore pursue parallel pathways, accepting that early deployments will rely on defenses with a shelf life measured in model generations.

Adversaries that combine learning with formal methods present a second threat, and such hybrid systems are now in production. The IRIS framework combines LLMs with static analysis and detects substantially more vulnerabilities than static analysis alone across large Java projects [19]. MoCQ autonomously generated vulnerability patterns that human analysts had missed [17]. Auto-Bug formalizes LLM-powered symbolic execution by decomposing whole-program reasoning into path-level sub-tasks [18]. These hybrid systems cross-validate neural predictions against formal semantics, reducing the effectiveness of defenses that rely solely on Statistical Opacity.

Each pathway fails when a core assumption is violated. Token fragmentation assumes subword tokenization trained on natural code distributions. Character-level and byte-level models render token fragmentation ineffective, and such tokenizer changes are increasingly likely. Semantic misdirection assumes models rely on statistical associations carried from pretraining. An adversary that fine-tunes specifically on opacity-transformed code may learn to discount misdirection patterns. An empirical question must be resolved early in the agenda. Does fine-tuning fully recover comprehension, or does misdirection impose a persistent penalty? Gradient poisoning assumes adversary pipelines ingest protected code without preprocessing that would strip the poison. Dead code elimination, a standard compiler optimization, removes many functionally neutral insertions. Effective poisoning must therefore use live constructs whose contribution to program output is negligible. The three pathways have disjoint failure modes. Defeating one does not defeat the others. Layered deployment ensures that circumventing any one defense requires a different strategy than circumventing the others. The cost of comprehensive defeat grows multiplicatively with the number of independent layers.

Individual protection is feasible with current techniques and requires only that a target codebase be transformed. Systemic protection requires opacity-transformed code to reach training pipelines at a scale large enough to degrade the resulting models. Production pipelines deduplicate aggressively, filter by quality signals, and weight sources by reliability. Opacity-transformed code exhibiting unusual identifiers, anomalous control flow, or inflated binary size may trigger quality filters and be excluded before reaching the training stage. If pipelines filter most transformed code, achieving systemic effect requires impractical levels of adoption. Transformations must therefore disrupt learned representations while preserving the surface statistics that pipeline filters check.

### 7.1 Illustrative Walkthrough

Our proof-of-concept implementation transforms a `Python` function, `below_zero`, that returns `True` if a running balance ever drops below zero. The function uses descriptive names, a `for` loop over an iterable, augmented assignment, and an early return, all idioms that a code LLM has seen many times in training. The result makes the gap between executability and learnability concrete. The prototype produces token fragmentation, semantic misdirection, and gradient poisoning, all of which are verified to pass the original test suite.

**Token fragmentation:** Every user-defined identifier outside the preserve set is replaced with a high-entropy token, and docstrings and type annotations are stripped in the same pass (Listing 1). Identifier names like `balance` carry a statistical signal in training corpora that `_w4ybpf` does not.

```python
def below_zero(_v9vjn4):
    _w4ybpf = 0
    for _uzqhqc in _v9vjn4:
        _w4ybpf += _uzqhqc
        if _w4ybpf < 0:
            return True
    return False
```

**Listing 1: Token fragmentation (`lexical` mode).**

```python
def below_zero(operations: List[int]) -> bool:
    balance = 0
    _it_0 = list(operations)
    _i_0 = 0
    while _i_0 < len(_it_0):
        op = _it_0[_i_0]
        balance = balance + op
        if balance < 0:
            return True
        _i_0 = _i_0 + 1
    return False
```

**Listing 2: Semantic misdirection (`syntactic` mode).**

**Semantic misdirection:** The `for`-over-iterable becomes a `while`-with-explicit-index, materializing a buffered `list(...)` call, and augmented assignment becomes explicit (Listing 2). Identifiers, the docstring, and type annotations are preserved. Only the keyword priors are violated. The canonical idioms occur with high frequency in training corpora, while the semantically equivalent rewrites occur far less often.

**Gradient poisoning:** Two control-flow constructs wrap the function body (Listing 3). The dead branch on $(2k^2+2k+1) \bmod 2 = 0$ is unreachable because the expression is odd for every integer $k$. The live wrapper on $(k^2 + k) \bmod 2 = 0$ always executes the body because $k(k + 1)$ is even. Both predicates evaluate $k$ as `int(bool(operations))`, which is 0 or 1 for any Python input type. The dead branch's tokens contribute to gradient updates during training, even though they have no effect on program behavior.

```python
def below_zero(operations: List[int]) -> bool:
    if (2 * int(bool(operations)) * int(bool(operations))
            + 2 * int(bool(operations)) + 1) % 2 == 0:
        return None
    if (int(bool(operations)) * int(bool(operations))
            + int(bool(operations))) % 2 == 0:
        balance = 0
        for op in operations:
            balance += op
            if balance < 0:
                return True
        return False
    else:
        pass
```

**Listing 3: Gradient poisoning (`opaque` mode).**

The Python interpreter executes the program according to its semantics and is independent of variable names, idiomatic choices, or dead branches. Code models predict tokens from surface patterns, so all three are relevant to prediction. The transformations are invisible to the interpreter and visible to the prediction surface that a code model relies on. Across the corpus, token fragmentation produces the largest per-token perplexity effect, gradient poisoning a smaller one, and semantic misdirection shows no measurable change. Composing all three yields a smaller uplift than token fragmentation alone, a non-monotonicity that holds across the corpus, because the semantic misdirection and gradient poisoning rewrites introduce token sequences common in training corpora, diluting the per-token contribution of the rare scrambled identifiers.

Paradigm transformation rewrites an algorithm in a different idiom while preserving its semantics. An imperative `below_zero` that uses a for loop, mutable accumulator, and early return becomes a functional composition that chains accumulate and any with neither loop nor mutation. Both forms compute the same predicate on the same inputs. Python training corpora are dominated by the imperative form, so the functional rewrite occupies a sparse region of code distributions. The prototype does not yet implement paradigm transformation because mechanical translation between paradigms requires recognizing what the algorithm computes, not just what its tokens say. Adaptive opacity differs from the other four pathways. It is not a single source-level transformation but a deployment-time strategy that dispatches different transformed versions of the function to different callers. A trusted caller receives the original, unaltered version. A suspected scraper receives a heavily transformed variant. Function name and output are unchanged for both, but the source code observed by the caller differs. The pathway treats the other four as primitives and depends on a runtime adversary-detection signal that is itself an open research problem rather than an engineering task.

## 8 Conclusion

Fifty years ago, the software engineering community faced a new threat from networked adversaries capable of intercepting communications. The response was cryptography, a technical primitive that assumed adversaries would see the data but ensured that seeing it yielded no useful information. Cryptography is now so fundamental that we cannot imagine secure systems without it. Today, we face an analogous threat from learning-based adversaries that extract patterns from code. Autonomous LLM-based agents already replicate complex attack chains, malware rewrites itself through foundation-model APIs, and traditional obfuscation no longer holds against neural comprehension. The response must be Statistical Opacity, a technical primitive that assumes adversaries will analyze our code but ensures that such analysis yields no useful capabilities. The code we write today trains the models of tomorrow. If we do not develop defenses, the most valuable software becomes training data for systems we do not control.

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
