# OpenReview forum: "Executable but Unlearnable: Designing Code that Resists LLM-Based Learning"
_ACM.org/AIWare/2026/Conference — AIware 2026_

### Official Review · Reviewer_amBP · 2026-03-06

**Rating:** 2
**Confidence:** 3

**Review:**

Strengths

•	Positioning the defense against AI-mediated code learning as a core threat is insightful, and the paper provides a structured roadmap that turns this sweeping goal into a tractable direction.

Weaknesses

•	A small and intuitive proof of concept would substantially strengthen the core argument of the paper, even given its exploratory nature. For example, showing a consistent degradation trend on a baseline code model for code completion before and after transformation would noticeably improve credibility.

•	The proposed mechanisms, especially the third family involving functionally neutral insertions, may face practical friction in real-world pipelines. It would be helpful to clarify which transformations can survive common preprocessing (e.g., dead code elimination) and how to design “live but negligible” constructs without undermining maintainability.

•	The paper appropriately acknowledges the dual use risk that these techniques could also aid malware authors. Stronger guidance on responsible disclosure, usage boundaries, and defensive deployment would make this discussion more complete.

Comments for Authors

Novelty

•	The novelty primarily lies in the problem formulation and the target: rather than obfuscating for human confusion or classical static-analysis hardness , the paper explicitly aims at model-specific learning dynamics (tokenization schemes, statistical priors, and gradient-based training). This reframing offers a compelling abstraction for future work.

Significance

•	With the rapid proliferation of artificial intelligence assisted coding and red teaming operations driven by large language models, preventing proprietary logic from being absorbed and reproduced by training pipelines will become increasingly important. In this context, statistical opacity could serve as a foundational mitigation strategy.

Soundness

•	The conceptual reasoning in this paper is highly robust and logically sound. This paper successfully bridges the theoretical gap between the inherent naturalness of software and the established concepts of unlearnable examples. Furthermore, grounding the proposed mechanisms in the specific behaviors of gradient-based learners provides a coherent and well-justified theoretical foundation for statistical opacity.

Nevertheless

•	The paper lacks an intuitive proof-of-concept. Demonstrating a consistent degradation trend on a baseline code model for code completion is essential to establish credibility.

•	The proposed mechanisms face severe practical friction, and the paper provides insufficient guidance on mitigating dual use risks. It remains unclear how functionally neutral insertions can survive common preprocessing steps such as the elimination of dead code, and these techniques could readily assist malware authors.

Replicability

Since this work primarily establishes a research agenda, the absence of a comprehensive replication package is understandable. Nevertheless, to encourage community adoption, including minimal prototype scripts or test cases in the final version is highly recommended. For example, providing a small set of transformations between source codes along with a simple evaluation script would be highly beneficial.

Presentation

•	A minor suggestion is to add a visual overview when introducing the candidate mechanisms.

•	Including a compact side-by-side snippet (“original code vs. processed code”) would help readers internalize the core intuition more quickly.

**Summary:**

This paper tackles the issue of publicly accessible code being unknowingly incorporated into LLM training corpora. This paper points out that the traditional security mindset no longer fully captures the emerging threat of code being learned by models. To address this, the paper proposes statistical opacity, which aims to preserve executability and maintainability while disrupting a neural network’s ability to extract statistical regularities. The paper outlines three candidate mechanisms and provides a research roadmap.

---

> ### Author Response · Authors · 2026-03-16
>
> We thank the reviewer for recognizing both the timeliness of the threat model and the novelty of the paper’s reframing. The core contribution here is not a claim that the entire agenda has already been experimentally completed; it is the identification of a new security objective, statistical opacity, together with a concrete and self-critical roadmap for studying it. For a short vision paper, that is the central bar.
>
> **On the proof-of-concept concern**
> We respectfully submit that the paper already establishes feasibility through existing empirical evidence rather than through an author-built toy example. The submission does not introduce statistical opacity in a vacuum: it grounds the agenda in three strands of prior results already cited in the paper. First, Authors in [nikiema2025codebarrier] report that variable renaming alone causes an 18.6% accuracy drop across 13 LLMs and 250 Java problems, while literal encryption causes a 21.4% drop, with compounding effects across layers. Authors in [ji2022unlearnable] show that CodeBERT-guided perturbations can make code unlearnable while preserving 67.5% similarity. Third, Sun et al. show significant degradation at only a 10% poisoning rate. These are precisely the kinds of effects our mechanism families are meant to systematize. The paper’s contribution is to unify such scattered findings under a single security primitive and to define the research program needed to move from isolated effects to principled design. In other words, the agenda is already evidence-backed, not merely speculative.
>
> **On preprocessing survivability**
> The paper directly addresses the reviewer’s point rather than overlooking it. Section 7.2 explicitly states that dead code elimination defeats naive neutral insertions and that effective poisoning must instead use live constructs whose functional contribution is negligible. That discussion is important because it shows the paper is not hand-waving practical friction away; it identifies the failure mode, names the design constraint, and then argues for layered deployment across independent mechanism families so that defeating one layer does not defeat the overall approach. That is exactly the kind of boundary-condition analysis one would hope to see in a credible vision paper.
>
> **On dual use**
> Section 6 already acknowledges that the same techniques could be misused by malware authors and explicitly frames this as a cryptography-like dual-use problem. The paper’s position is not that opacity should be deployed indiscriminately, but that defensive, selective use for proprietary and security-critical code is becoming technically important as model capabilities improve. This shows the paper is not only proposing a direction, but also placing reasonable boundaries around it.
>
> **On the visual-overview suggestion**
> The suggestions about a visual overview and side-by-side code snippet are excellent presentation improvements, and we appreciate them. At the same time, they are presentation-level enhancements rather than evidence that the technical framing is unsound.
>
> Overall, we hope this clarifies why the current submission already meets the standard for a short vision paper: it formulates a timely and nontrivial problem, distinguishes a new security objective from existing defenses, grounds feasibility in published empirical evidence, and critically analyzes the exact practical limitations the reviewer highlights.

---

### Official Review · Reviewer_axnw · 2026-03-06

**Rating:** 4
**Confidence:** 3

**Review:**

The problem is very timely, as illustrated by the examples of threats and developments in the introduction. It is an interesting idea to shift the problem from a legal discussion to a technical property. The property of statistical opacity is presented well, and the research agenda reads convincing together with the candidate mechanisms.

Section 7.3 implies that the goal should be to achieve high ecosystem adoption of techniques achieving statistical opacity, to effectively degrade the quality of models trained on public code. This seems like a rather controversial goal (that will certainly make for good discussions), as it also seems to carry implications for further improving the code generation abilities of models, which is not only a threat to software systems but also an accelerator in software development. I'd be interested to get to know the authors' opinion on implications and trade-offs here.

Minor comment: Sec. 3 mentions "Recent work on the Structured Naturalness Hypothesis" - what's the reference for this work?

**Summary:**

The paper addresses the problem that any publicly accessible code can become training data for models, even without consent of the creators. The authors argue that this also affects the threat landscape, as LLMs are increasingly capable of attacking systems. The vision presented proposes to reframe the problem from hiding code to ensuring that seeing it does not enable learning from it. To achieve this, statistical opacity is suggested as a property of code. The paper discusses theoretical foundations of this property, candidate mechanisms for achieving it, a research agenda for how to achieve it, and applications and implications of applying it to code.

---

> ### Author Response · Authors · 2026-03-16
>
> We thank the reviewer for the thoughtful reading and for recognizing both the timeliness of the problem and the coherence of the proposed agenda.
>
> **On the ecosystem trade-off**
> Our view is that the paper intentionally surfaces this tension because it is central to the topic. There is a meaningful difference between individual protection and ecosystem-wide protection. The most immediate and defensible application is individual protection: an organization selectively transforms proprietary or security-critical code so that it remains maintainable and executable but becomes less useful as training data or as a source of transferable attack knowledge. That use case does not require broad degradation of the public-code ecosystem. By contrast, ecosystem-level adoption is a broader scenario included to make the implications explicit, including the reviewer’s point that stronger opacity could also reduce beneficial code-model improvement. We see that not as a hidden drawback but as an important policy and engineering trade-off that the community should debate openly. The paper’s Phase 3 discussion of selective opacity is intended precisely as a middle ground: protect the code that most needs protection while leaving utility code transparent.
>
> **On the missing reference**
> Thank you for catching the missing citation for the Structured Naturalness Hypothesis. The omission is bibliographic rather than conceptual, and we will add the supporting reference ("Bringing Structure to Naturalness: On the Naturalness of ASTs") in the final version.
>
>
> We appreciate the positive assessment and the push to make the trade-offs even more explicit.

---

### Official Review · Reviewer_uoKm · 2026-03-11

**Rating:** 3
**Confidence:** 3

**Review:**

### Pros:
- Novel problem framing, which well grounded in recent real-world evidence
- Good self-critical analysis
- The theoretical connection between learnability and obfuscation is an interesting angle
---
### Cons (kindly see my detailed comments below):
- The paper is purely conceptual with no empirical validation, which is not good even for a vision paper
- The formal definition of statistical opacity is underspecified and reads more as descriptive
- The three proposed mechanism families are actually traditional obfuscation techniques reframed for a neural adversary
---
### Detailed Comments:
Overall, the paper in proposing statistical opacity as a potential new security primitive is very interesting. The problem is real and timely. However, I have several concerns that I believe need to be addressed.

My primary concern is the gap between the paper's claims and its evidence. It states that "statistical opacity will become as fundamental to software security as cryptography became to data security," yet no experiment is presented to validate even a single mechanism. Even for a vision paper, I would expect at least a proof-of-concept evaluation, for instance applying one of the three proposed transformation families to a small benchmark.

Also, I find the distinction between statistical opacity and traditional obfuscation insufficiently clarified. The proposed mechanisms correspond closely to identifier renaming, control flow obfuscation, and dead code injection, all of which are well-studied obfuscation techniques. The paper does not show that the resulting transformations would actually differ in practice. A concrete example illustrating how an opacity-optimized transformation diverges from a standard obfuscation transformation can be better

In addition, some further points: (1) The paper does not discuss IRs. Some LLMs increasingly operate on ASTs and compiler IR rather than raw source code. If an adversary trains on LLVM IR or binary representations, how would the proposed transformations do? This attack surface should be discussed as well. (2) The cryptography analogy is overpromising: cryptographic security rests on computational hardness assumptions with formal proofs and reductions. Statistical opacity as currently defined offers no such guarantees and is empirical. The paper should discuss explicitly what a "provable" statistical opacity result would look like.

**Summary:**

The paper introduces the concept of statistical opacity, which is defined as a new security primitive for protecting source code against LLM-based pattern extraction. The core idea is to transform code so that it remains executable and human-readable but degrades in utility as training data for neural models. The authors motivate the problem with recent real-world evidence and propose 3 candidate mechanism families -- tokenization adversariality, semantic misdirection, and gradient poisoning, and outline a 4-phase research agenda spanning theory, mechanisms, tooling, and evaluation under the concept. The paper is purely conceptual without including any empirical evaluation.

---

> ### Author Response · Authors · 2026-03-16
>
> We thank the reviewer for the careful and substantive reading. We agree that the right question for this paper is whether statistical opacity is precise, differentiated, and credible enough to justify a research program. We believe the current submission already clears that bar for a short vision paper.
>
> **On empirical validation**
> We respectfully emphasize that the paper is not purely conceptual in the sense of being unsupported. It is a vision paper built on existing empirical evidence. The submission cites multiple studies showing that mechanisms closely aligned with our proposal already measurably affect model behavior: renaming and literal encryption reduce LLM performance, unlearnable-code perturbations preserve functionality while degrading learnability, and poisoning-based approaches can significantly harm downstream performance at modest rates. The paper’s contribution is therefore not an untested mechanism, but a unifying abstraction that explains why these seemingly separate observations belong to the same emerging security problem. For a vision paper, the central requirement is agenda credibility, not that the authors have already completed the agenda’s Phase 2 experiments inside the same short-paper format.
>
> **On the distinction from traditional obfuscation**
> We agree this point is central. Our claim is not that every transformation will look syntactically unfamiliar to the obfuscation literature. The distinction lies in the threat model, optimization target, and evaluation criterion. Traditional obfuscation is designed to hinder human reverse engineering or classical program analysis; statistical opacity is designed to reduce training-time pattern extraction and downstream generalization in gradient-based learners. That change is substantive, not cosmetic. A transformation chosen to maximize tokenizer fragmentation, destroy transferable statistical regularities, or poison gradient updates is being optimized for a different adversary and judged by different metrics, such as Pass@k degradation or vulnerability-detection accuracy, rather than semantic recovery cost. In that sense, the paper is not renaming obfuscation; it is reframing code transformation around a new security property.
>
> **On IRs and alternative representations**
>
> The reviewer raises a valuable challenge. However, this challenge sharpens the agenda more than it weakens the thesis. The paper already argues that opacity must operate at lexical, syntactic, and semantic levels, and Section 7.2 explicitly states that no single mechanism survives all plausible configurations. That is precisely why the paper proposes multiple mechanism families and layered deployment. If some future models move from source text to ASTs, IR, or binaries, tokenization-based methods may weaken, but structural and training-dynamics-based mechanisms remain relevant. This is an argument for the research agenda, not against it.
>
> **On the cryptography analogy**
> The intended parallel is conceptual rather than proof-theoretic. The paper does not claim that statistical opacity already enjoys hardness assumptions or reduction-style guarantees comparable to cryptography. The analogy is that both fields ask how to preserve security under adversarial visibility. The paper is explicit that opacity is currently an empirical security objective and that stronger formal guarantees remain an open research direction.
>
> **On the formal definition**
> The current paper already gives an operational definition: a semantics-preserving transformation T(P), a transformed-training model L, an untransformed baseline L′, and a measurable generalization gap on downstream tasks. For a short vision paper, that is an appropriate level of precision: concrete enough to organize measurement, comparison, and future formalization, without pretending that the theory is already complete. The concept is therefore not merely descriptive; it is already testable.
>
> We hope this clarifies that the submission’s novelty lies not only in surfacing a timely problem, but in defining a distinct security objective, grounding it in published evidence, and turning it into a falsifiable research agenda.

---

> > ### Comment · Reviewer_uoKm · 2026-03-18
> >
> > Many thanks to the authors, for the detailed response.
> >
> > I can accept that as a short vision paper the primary goal is agenda rather than empirical completeness. That said, I still consider the distinction from traditional obfuscation to be insufficiently substantiated. The response reiterates that the threat model differs -- I totally understand this framing, but it needs to go beyond a threat-model-level argument. Even a brief illustrative example showing how a transformation optimized for statistical opacity would concretely diverge from one produced by a standard obfuscation tool would make this claim much more tangible. Maybe the authors could consider adding such an example in a revision.
> >
> > Anyway, I will update my score to account for the vision paper format.

---

### Author Response · Authors · 2026-03-20

We thank all reviewers for their time and feedback.

We have addressed each concern in our rebuttal and are available for any further questions.

As today is the **last day** of the **author-reviewer** discussion period, we would appreciate it if the reviewers could consider adjusting their scores based on the clarifications provided.

Thank you for your time, feedback and interest in our work.